# Immunogenicity and safety of Quadrivalent Influenza HA vaccine compared with Trivalent Influenza HA vaccine and evaluation of Quadrivalent Influenza HA vaccine batch-to-batch consistency in Indonesian children and adults

Eddy Fadlyana[1]*, Meita Dhamayanti[1], Rodman Tarigan[1], Susantina Prodjosoewojo[2], Andri Reza Rahmadi[2], Rini Mulia Sari[3], Kusnandi Rusmil[1], Cissy B. Kartasasmita[1]

1 Faculty of Medicine Universitas Padjadjaran/Hasan Sadikin Hospital, Department of Child Health, Bandung, Indonesia, 2 Faculty of Medicine Universitas Padjadjaran/Hasan Sadikin Hospital, Department of Internal Medicine, Bandung, Indonesia, 3 Surveillance and Clinical Trial Division, PT Bio Farma, Bandung, Indonesia

* edfadlyana@gmail.com

## Abstract

One of the newest strategies developed by the Global Influenza Strategy has been to broaden the composition of the current influenza vaccine formulations from trivalent products to quadrivalent products. This study aimed to assess the immunogenicity and safety of Quadrivalent Influenza HA vaccine (QIV) compared with Trivalent Influenza HA vaccine (TIV) and to evaluate three consecutive batches of QIV equivalence in Indonesian children and adults. This was an experimental, randomized, double blind, four arm parallel group bridging study involving unprimed healthy children and adults aged 9–40 years. A total of 540 subjects were enrolled in this study and randomized into four arm groups. Each subject received one dose of TIV or QIV with three different batch codes. Serology tests were performed at baseline and 28 days after vaccination. Hemagglutination inhibition (HI) antibody titers were analyzed for Geometric Mean Titer (GMT), seroprotection, and seroconversion rates. Solicited, unsolicited, and serious adverse events were observed up to 28 days after vaccination. A total of 537 subjects completed the study per protocol and were analyzed for immunogenicity criteria. All randomized subjects were analyzed for safety criteria. The percentage of the subjects with anti-HI titer $\geq$1:40 28 days after QIV vaccination was 99.5% for A/H1N1; 99.5% for A/H3N2; 93.1% for B/Texas; and 99.0% for B/Phuket. The seroprotection, GMT, and seroconversion rates of QIV were not significantly different from those of TIV for the common vaccine strains ($p > 0.01$) and were significantly different from those of TIV for the added B/Phuket strains ($p < 0.01$). Most solicited injection-site and systemic reactions with either vaccine were mild to moderate and resolved within a few days. Antibody response to QIV were equivalence among vaccine batches and comparable between age groups for each of the 4 strains. QIV was immunogenic and well-tolerated and had

**Data Availability Statement:** All relevant data are within the paper.

**Funding:** YES - Funding for this trial was provided by PT Bio Farma Indonesia, no 04310/DIR/XI/2017, PO-00017403. PT Bio Farma Indonesia was involved in the study design, data collection, data analysis and preparation of the manuscript.

**Competing interests:** Novilia Sjafri Bachtiar and Rini Mulia Sari are employees of PT Bio Farma at the time of the conduct of this study and manuscript preparation. PT Bio Farma Indonesia was involved in the study design, data collection, data analysis and preparation of the manuscript. This does not alter our adherence to PLOS ONE policies on sharing data and materials.

immunogenicity and safety profiles compared with TIV for all common strains. The immunogenicity of the three batches of QIV was equivalent for the four strains.

**Trial registration. Clinical Trial registration**: NCT03336593.

## Introduction

Influenza is an acute viral illness of the respiratory tract and poses a substantial public health burden in terms of morbidity, mortality, and costs. The World Health Organization (WHO) reported that 3–5 million cases of severe influenza occur each year worldwide, resulting in about 290,000 to 650,000 related deaths per year. One of the goals of the Global Influenza Strategy for 2019–2030 is to reduce the burden of seasonal influenza by promoting research and innovation for improved influenza vaccines [1]. Annual influenza immunization is recommended in elderly subjects, children aged six months or more, pregnant women, and individuals with chronic conditions, such as respiratory/heart/liver diseases, diabetes, or a weakened immune system. These categories are at heightened risk of influenza-related complications and mortality [2].

Internationally available vaccines for controlling seasonal influenza are safe and effective and have the potential to prevent significant annual morbidity and mortality. The WHO annually recommends composing vaccines based on global virological surveillance. Annual Trivalent influenza HA vaccines (TIVs) contain two influenza A strains (H1N1 and H3N2) and only one influenza B virus [3]. Therefore, the effectiveness of TIVs depends on the degree of matching between the vaccine strain and circulating viral strains. In the last two decades, four major and at least eight minor mismatches between vaccine and circulating B viruses have occurred in the northern hemisphere, thus impairing the performance of TIVs [4]. Specifically, Ambrose et al. observed that a B-mismatch between vaccine and circulating strains occurred in Europe in 5 of 10 seasons between 2001 and 2011 [5].

In February 2009, the Food and Drug Administration (FDA), considered in adding an additional influenza B strain in the antigenic composition of seasonal influenza vaccines [6]. In 2012, WHO recommended to include both expected B strains in the vaccine composition of Quadrivalent Influenza HA vaccines (QIVs) for seasonal immunization [7,8].

At first, Bio Farma formulated TIVs and have conducted several studies on seasonal TIVs between 2008 and 2014. The results of these studies were consistent. TIVs were well-tolerated and induced high antibody titer against influenza antigens and no serious adverse events (AEs) during the study [9–13]. In 2016, Bio Farma starts to formulate the quadrivalent inactivated influenza vaccine that could potentially provide wider protection against influenza B viruses. Several countries has been conducted the studies using Influenza HA Vaccine (quadrivalent vaccine), but none from Indonesia [8,14,15]. This was the first QIV study conducted by Bio Farma on subjects aged 9–40 years in Indonesia. This study aimed to assess the immunogenicity and safety of QIV compared with TIV and to evaluate batch-to-batch consistency in three consecutive batches of QIV.

## Materials and methods

### Study design

This was an experimental, randomized, double blind, four arm parallel group bridging study to assess immune response and safety after one dose of quadrivalent and trivalent influenza

HA vaccine in subjects 9–40 years old in clinically health children and adults aged 9–40 years. The study was a collaboration between the Department of Child Health, Faculty of Medicine, Universitas Padjadjaran, and PT Bio Farma (Persero), Indonesia. This study was approved by the Research Ethics Committee of the Faculty of Medicine, Universitas Padjadjaran, and conducted in accordance with the Declaration of Helsinki and the International Conference on Harmonization Good Clinical Practice guidelines. Written informed consent was obtained from the participants or their parents before performing any study-specific procedure.

## Study subjects

A total of 540 subjects were enrolled in this study. Subjects were enrolled from 3 primary care centers in Bandung City, Ibrahim Adjie Primary Health Center, Puter Primary Health Center, and Garuda Primary Health Center, from October 2017 to June 2018. The primary inclusion criteria included healthy children and adults aged 9–40 years who have signed the informed consent form and committed to complying with study instructions and trial schedules. Subjects were not eligible if the subject have enrolled or scheduled to be enrolled in another trial, presented with mild, moderate, or severe illness with a fever (axillary temperature ≥37.5°C), history of allergy to egg, chicken protein, or other vaccine, uncontrolled coagulopathy or blood disorders contraindicating intramuscular injection, received in the previous 4 weeks a treatment likely to alter the immune response [intravenous immunoglobulins, blood-derived products, or long-term corticosteroid therapy (>2 weeks)], pregnancy and lactation (adult), have any abnormality or chronic disease which according to the investigator might interfere with the assessment of the trial objectives, have already immunized with influenza vaccine within 1 year or any vaccination within 1 month before and after immunization of Quadrivalent Influenza Vaccine.

## Randomization and blinding

For each subject recruited, the inclusion number was allocated in the chronological order of the subject, which was included in the trial from I-001 to I-180 (for the 9–12-year age group), II-001 to II-180 (for the 13–17-year age group), and III-001 to III-180 (for the 18–40-year age group). The subjects were randomized into treatment groups. The doctor strictly followed the list of randomization provided by Bio Farma. Treatment was allocated in accordance with a randomization list so that each randomization number corresponded to only one strictly randomly assigned treatment group (QIV batch A, QIV batch B, QIV batch C, and TIV).

## Vaccines and vaccination schedule

The QIV vaccine was formulated by PT Bio Farma (Persero), Indonesia, using bulks imported from Japan. The investigational QIV contained 15 µg HA from each of 4 strains, A/California/7/2009 (X-179A) (H1N1) pdm09n, A/Hong Kong/4801/2014 (X-263) (H3N2), B/Texas/2/2013, and B/Phuket/3073/2013, in a 0.5 ml dose, with batch numbers A: 3070117, B: 3070217, and C: 3070317. TIV contained 15 µg HA of each of 3 strains, A/California/7/2009 (X-179A) (H1N1) pdm09n, A/Hong Kong/4801/2014 (X-263) (H3N2), and B/Texas/2/2013, in 0.5 ml dose. Each subject received one dose (0.5 ml) of TIV or QIV with different batch numbers: 3070117, 3070217, and 3070317 for batches A, B, and C, respectively, according to the randomization.

## Sample size

The sample size was determined based on a 95% confidence interval (CI) and a test power of 80%. The required sample size was 112 in each batches, with the assumption that not all the subjects could complete the study, the total number of subjects was added at least 20% from the minimum requirement (N x 1.2) = 134. Approximately 135 subjects per group will be involved in this study.

## Study analysis

Demographic data were expressed as mean, standard deviation, and range values. Analysis of Geometric Mean Titer (GMT), seroprotection, and seroconversion rates between the vaccine groups was performed using Kruskal-Wallis and Chi-square or Kolmogorov-Smirnov tests. Values of $p < 0.05$ indicated statistically significant differences between groups. Immunogenicity analyses were performed on the per protocol population. Safety data were analysed in the intention to treat population.

The safety analyses were based on the intention-to-treat population analyses. The safety data were collected up to 28 days after the vaccination. The subjects were provided with a diary card to record the appearance, duration, and intensity (mild, moderate, or severe) of any solicited AE (local pain, redness, swelling, induration, fever, fatigue, and myalgia) and unsolicited AE. Local pain was graded as mild (mild pain at the injection-site when touched), moderate (pain with movements), and severe (significant pain at rest). Redness, induration, and swelling intensity were measured using a plastic bangle and categorized as mild ($<5$ cm), moderate (5–10 cm), and severe ($>10$ cm). Fever was graded as mild (38.0˚C–38.4˚C), moderate (38.5˚C–38.9˚C), and severe ($\geq$39.0˚C). Fatigue, myalgia, and unsolicited events were graded as mild (no interference with activity), moderate (some interference with activity), and severe (prevents daily activity, requires medical intervention).

The primary outcome was to evaluate the percentage of subjects with anti-HI titer $\geq$ 1:40, 28 days after Influeza HA vaccination. The secondary outcome was to evaluate geometric mean titres, percentage of subjects with increasing antibody titer $\geq$ 4 times and/or percentage of subjects with transition of seronegative to seropositive between one dose of quadrivalent and trivalent influenza HA vaccine in subjects 9–40 years old and between each batch number of Quadrivalent Influenza HA vaccine and the incidence rate and intensity of adverse events or any serious adverse events within 30 minutes, 72 hour and 28 days after immunization.

## Results

The demographic characteristics of study participants showed a fair distribution in gender and age (Table 1).

**Table 1. Demographic characteristics of study participants.**

| Charateristics | Batch A (n = 135) | Batch B (n = 135) | Batch C (n = 135) | TIV (n = 135) |
|---|---|---|---|---|
| Sex, n (%) | | | | |
| Male | 67 (49.6) | 67 (49.6) | 57 (42.2) | 64 (47.4) |
| Female | 68 (50.4) | 68 (50.4) | 78 (57.8) | 71 (52.6) |
| Age (y) | | | | |
| Mean ± standard deviation | 17.21 ± 7.64 | 17.70 ± 8.48 | 17.74 ± 8.31 | 17.18 ± 8.27 |
| Median | 15 | 14 | 14 | 14 |
| Range | 9–39 | 9–40 | 9–40 | 9–39 |

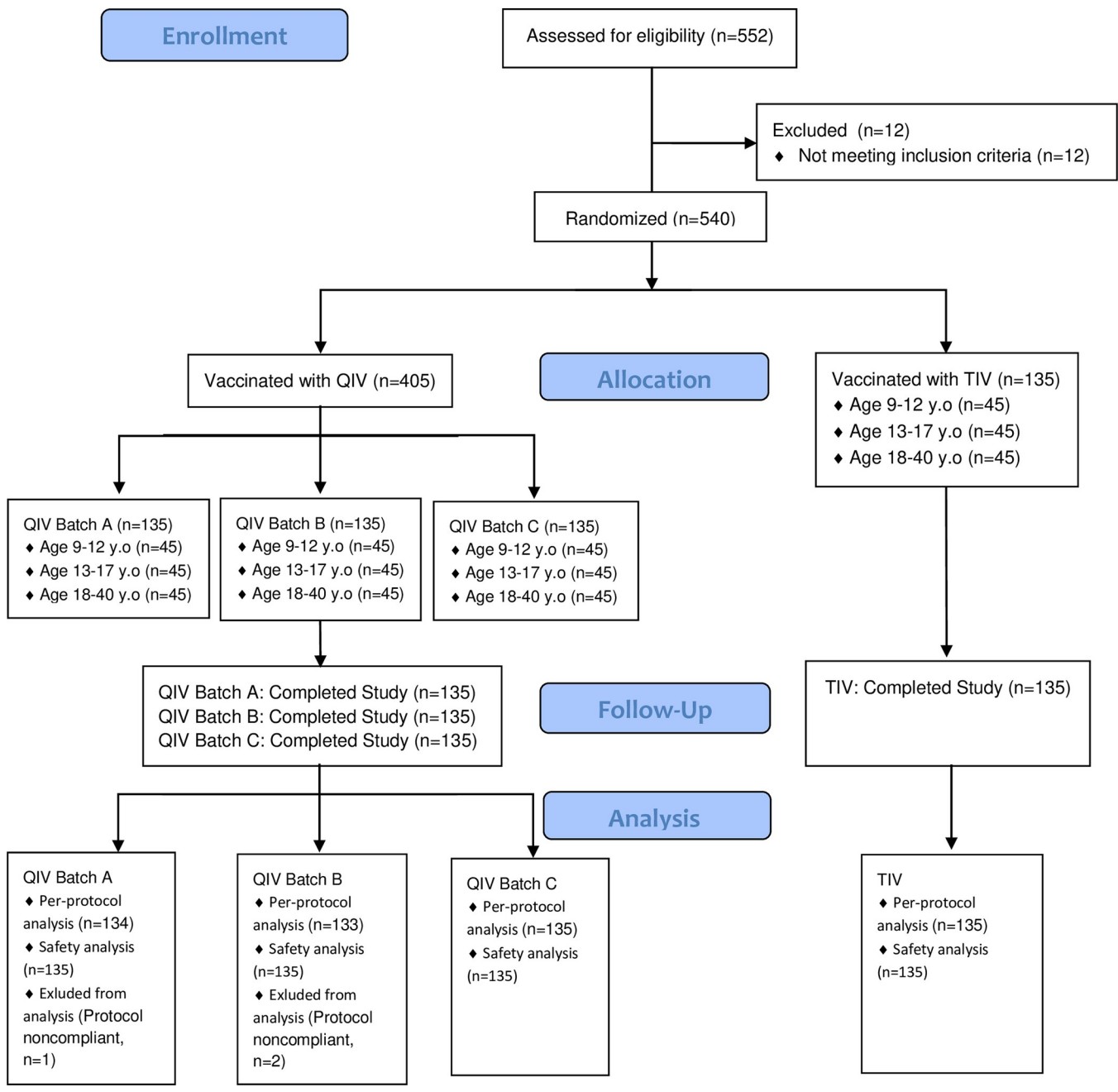

**Fig 1. Flow Chart for the participants enrollment.**

In this study, 405 and 135 subjects were enrolled in the QIV (Batch A, B and C) and TIV groups, respectively. Of the 405 subjects in the QIV group, three were excluded from immunogenicity analysis because of protocol non-compliance (Fig 1).

Comparison of the percentage of subjects with anti-HI titer ≥1:40 28 days after QIV and TIV was shown on Table 2. No difference in seroprotection was observed between one dose of QIV and TIV, except for B/Phuket/3073/2013.

Comparison of GMT between subjects who received QIV and TIV was shown on Table 3 and Fig 2. No significant differences in GMT 28 days after immunization between one dose of

**Table 2. Influenza seroprotection rate before and 28 days after immunization.**

| Strain | Pre-immunization | | p* | Post-immunization | | p* |
|---|---|---|---|---|---|---|
| | QIV (n = 402) | TIV (n = 135) | | QIV (n = 402) | TIV (n = 135) | |
| A/H1N1 ≥1:40 HI, n | 274 | 82 | 0.115 | 400 | 133 | 0.264 |
| % (95% CI) | 68.2 (63.4–72.5) | 60.7 (52.5–69.0) | | 99.5 (98.2–99.9) | 98.5 (94.8–99.6) | |
| A/H3N2 ≥1:40 HI, n | 267 | 92 | 0.712 | 400 | 132 | 0.104 |
| % (95% CI) | 66.4 (61.7–70.9) | 68.1 (60.3–76.0) | | 99.5 (98.2–99.9) | 97.8 (93.7–99.2) | |
| B/Texas ≥1:40 HI, n | 146 | 53 | 0.540 | 374 | 124 | 0.647 |
| % (95% CI) | 36.3 (31.8–41.4) | 39.3 (31.0–47.5) | | 93.0 (90.1–95.1) | 91.9 (86.0–95.4) | |
| B/Phuket ≥1:40 HI, n | 249 | 76 | 0.246 | 398 | 123 | <0,001 |
| % (95% CI) | 61.9 (57.1–66.6) | 56.3 (47.9–64.7) | | 99.0 (97.5–99.6) | 91.1 (85.1–94.8) | |

*Chi-square test.

QIV and TIV in the 9–40-year-old subjects for A/California/7/2009 (X-179A) (H1N1) pdm09, A/Hong Kong/4801/2014 (X-263) (H3N2), and B/Texas/2/2013 ($p = 0.322$, $p = 0.536$, and $p = 0.378$, respectively). However, in strain B/Phuket/3073/2013, GMT 28 days after immunization was significantly higher in QIV group than in TIV group ($p < 0.001$).

The transition from seronegative to seropositive is defined as a pre-vaccination titer <1:40 HI units and a post-vaccination titer ≥1:40 HI units. Seroconversion was also defined as increasing antibody titer ≥4 times and transition from seronegative to seropositive.

Table 4 shows there are no significant differences of seroconversion and transition from seronegative to seropositive between one dose of QIV and TIV in the 9–40-year-old subjects for A/California/7/2009 (X-179A) (H1N1) pdm09, A/Hong Kong/4801/2014 (X-263) (H3N2), and B/Texas/2/2013 ($p = >0.05$), except for B/Phuket/3073/2013, where QIV group was significantly higher than TIV group in the seroconversion and transition from seronegative to seropositive ($p < 0.001$).

Batch-to-batch equivalence for each strain was concluded if the two-sided 95% CI of each strain GMT ratio of the compared batches was between 0.67 and 1.5 (Table 5) [16]. The GMT ratios of the overall post-vaccination GMTs for each pair of lots for each strain were all between 0.67 and 1.5.

**Table 3. Geometric Mean Titers (GMT) to influenza A (H1N1), A (H3N2), and B virus strains Pre- and 28 days post-immunization.**

| Strain | Pre-immunization | | P*) | Post-immunization | | P*) |
|---|---|---|---|---|---|---|
| | QIV | TIV | | QIV | TIV | |
| A/California/7/2009/H1N1 GMT/95%CI | 57.12 (51.04–63.93) | 49.35 (41.25–59.05) | 0.250 | 910.75 (833.68–944.72) | 939.07 (777.68–1133.7) | 0.322 |
| A/Hongkong/4801/2014/H3N2 GMT/95%CI | 51.67 (46.75–57.10) | 52.95 (44.67–62.78) | 0.701 | 502.81 (453.42–557.44) | 554.75 (467.52–658.11) | 0.536 |
| B/Texas/2/2013 GMT/95%CI | 26.34 (24.17–28.71) | 25.47 (22.20–29.21) | 0.853 | 330.37 (292.15–373.51) | 358.26 (281.77–455.41) | 0.378 |
| B/ Phuket/3073/2013 GMT/95%CI | 42.87 (39.32–46.75) | 42.54 (36.06–50.19) | 0.642 | 348.42 (317.76–382.12) | 112.85 (94.65–134.56) | <0.001 |

*) Mann-Whitney test.

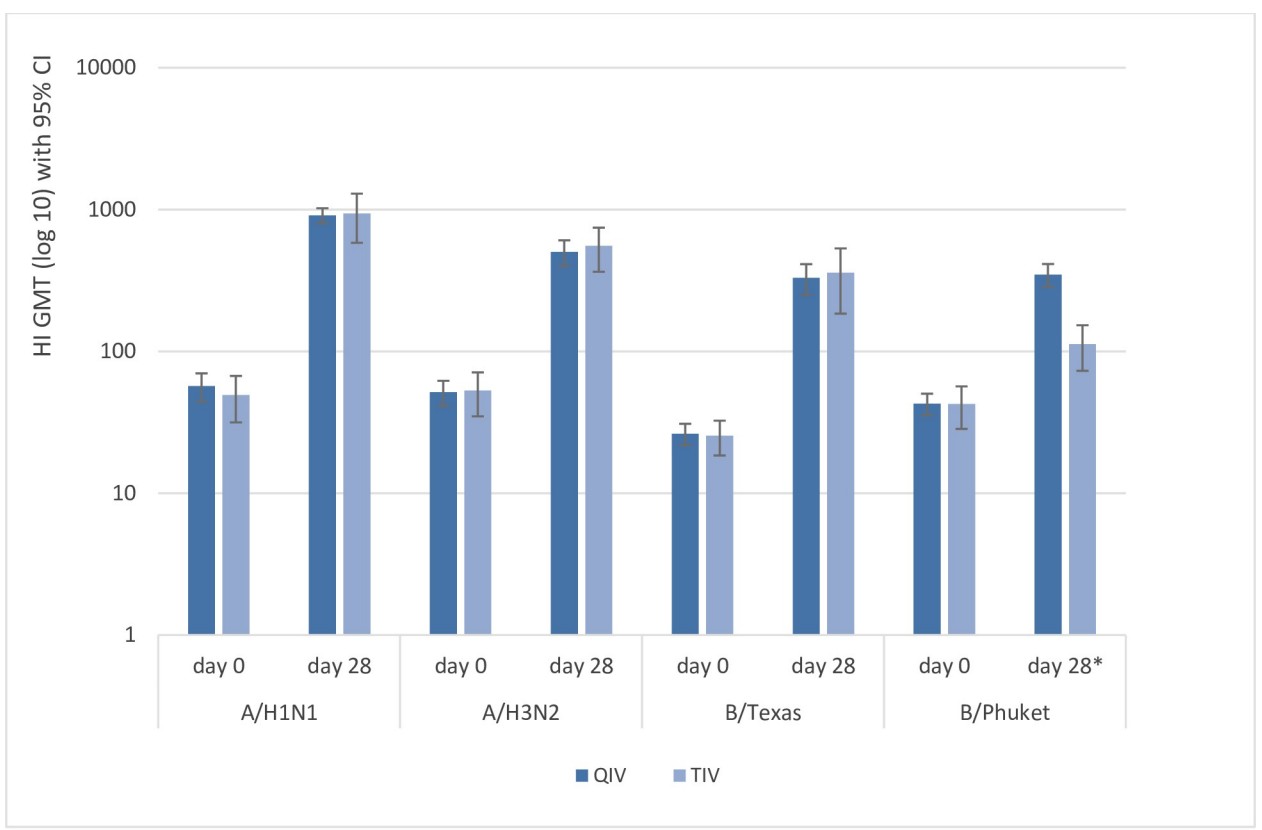

*p<0.001

**Fig 2. Hemagglutination inhibition (HI) Geometric Mean Titers (GMT) before and 28 days after QIV and TIV immunization.**

Solicited and unsolicited post-vaccination AEs were categorized as immediate (within 30 min), intermediate (30 min to 72 h), and delayed (72 h to 28 days) reactions (Figs 3 and 4).

Most local and systemic AEs reported in the QIV and TIV groups had mild intensity. Mild pain was the most local adverse reaction, which occurred in 15.3% and 17.8% of the subjects in

**Table 4. Differences in seroconversion rates.**

| Strain | Increasing antibody titer ≥ 4 times [a] n (%) | | | | Transition from seronegative to seropositive[b] (%)[c] | | | |
|---|---|---|---|---|---|---|---|---|
| | QIV | TIV | %Diff (95% CI) | p | QIV | TIV | Diff (95% CI) | P[d] |
| A/H1N1) | 333 (82.2) | 115 (85.2) | -3.0 (-9.4; 4.8) | 0.525 | 98.4% | 96.2% | 2.2 (-0.8; 6.7) | 0.582 |
| A/ H3N2 | 331 (81.7) | 113 (83.1) | -1.4 (-8.1; 6.6) | 0.717 | 98.5% | 93.0% | 5.5 (1.6; 10.8) | 0.092 |
| B/Texas | 349 (86.1) | 114 (84.4) | 1.7 (-4.6; 9.4) | 0.489 | 89.1% | 86.6% | 2.5 (-3.4; 9.7) | 0.541 |
| B/Phuket | 335 (82.7) | 56 (41.5) | 41.2 (31.9; 49.9) | <0.001 | 97.4% | 79.7% | 17.7 (11.0; 24.9) | <0.001 |

CI, confidence interval.

[a] Number of subjects (*n*) on increasing antibody titer for group QIV = 402 and group TIV = 135.

[b] Number of subjects (*n*) on transition from seronegative to seropositive for each group, and each strain was based on the number of seronegative subjects at baseline (pre-vaccination).

[c] Percentage (%) defined as the percentage of subjects with anti-HI titer <1:40 HI (seronegative) at baseline and ≥1:40 HI (seropositive) after vaccination.

[d] *p* value based on the Fisher exact test.

**Table 5. QIV batch-to-batch comparison.**

| Age group | Strain | Comparison | GMT post-vaccination ratio (95% CI) | Equivalence |
|---|---|---|---|---|
| 9–12 years | A/H1N1 | Batch A vs. batch B | 1.030 (0.996–1.061) | Yes |
| n = 135 | | Batch A vs. batch C | 1.020 (0.976–1.072) | Yes |
| | | Batch B vs. batch C | 1.082 (0.988–1.176) | Yes |
| | A/H3N2 | Batch A vs. batch B | 1.074 (0.996–1.152) | Yes |
| | | Batch A vs. batch C | 1.027 (0.971–1.083) | Yes |
| | | Batch B vs. batch C | 0.987 (0.928–1.045) | Yes |
| | B/Texas | Batch A vs. batch B | 0.951(0.941–1.061) | Yes |
| | | Batch A vs. batch C | 1.003 (0.867–1.138) | Yes |
| | | Batch B vs. batch C | 1.107 (0.979–1.236) | Yes |
| | B/Phuket | Batch A vs. batch B | 1.076 (1.004–1.147) | Yes |
| | | Batch A vs. batch C | 1.089 (1.019–1.159) | Yes |
| | | Batch B vs. batch C | 1.045 (0.960–1.129) | Yes |
| 13–17 years | A/H1N1 | Batch A vs. batch B | 1.010 (0.958–1.063) | Yes |
| n = 132 | | Batch A vs. batch C | 0.981 (0.938–1.023) | Yes |
| | | Batch B vs. batch C | 0.982 (0.944–1.021) | Yes |
| | A/H3N2 | Batch A vs. batch B | 0.997 (0.925–1.068) | Yes |
| | | Batch A vs. batch C | 1.023 (0.949–1.096) | Yes |
| | | Batch B vs. batch C | 1.053 (0.984–1.121) | Yes |
| | B/Texas | Batch A vs. batch B | 1.025 (0.916–1.133) | Yes |
| | | Batch A vs. batch C | 0.972 (0.881–1.064) | Yes |
| | | Batch B vs. batch C | 1.000 (0.913–1.087) | Yes |
| | B/Phuket | Batch A vs. batch B | 1.009 (0.948–1.070) | Yes |
| | | Batch A vs. batch C | 0.971 (0.909–1.034) | Yes |
| | | Batch B vs. batch C | 0.984 (0.913–1.055) | Yes |
| 18–40 years | A/H1N1 | Batch A vs. batch B | 1.051 (0.982–1.120) | Yes |
| n = 135 | | Batch A vs. batch C | 1.073 (0.996–1.150) | Yes |
| | | Batch B vs. batch C | 1.036 (0.975–1.097) | Yes |
| | A/H3N2 | Batch A vs. batch B | 1.123 (1.034–1.211) | Yes |
| | | Batch A vs. batch C | 1.062 (0.968–1.155) | Yes |
| | | Batch B vs. batch C | 0.985 (0.889–1.081) | Yes |
| | B/Texas | Batch A vs. batch B | 1.028 (0.927–1.129) | Yes |
| | | Batch A vs. batch C | 1.064 (0.958–1.169) | Yes |
| | | Batch B vs. batch C | 1.067 (0.969–1.165) | Yes |
| | B/Phuket | Batch A vs. batch B | 1.065 (0.987–1.144) | Yes |
| | | Batch A vs. batch C | 1.084 (0.998–1.170) | Yes |
| | | Batch B vs. batch C | 1.042 (0.967–1.117) | Yes |
| All age (9–40 years) | A/H1N1 | Batch A vs. batch B | 1.029 (0.996–1.061) | Yes |
| n = 402 | | Batch A vs. batch C | 1.046 (1.003–1.089) | Yes |
| | | Batch B vs. batch C | 1.030 (0.989–1.070) | Yes |
| | A/H3N2 | Batch A vs. batch B | 1.066 (1.020–1.111) | Yes |
| | | Batch A vs. batch C | 1.037 (0.994–1.080) | Yes |
| | | Batch B vs. batch C | 1.008 (0.964–1.051) | Yes |
| | B/Texas | Batch A vs. batch B | 1.001 (0.941–1.061) | Yes |
| | | Batch A vs. batch C | 1.014 (0.950–1.078) | Yes |
| | | Batch B vs. batch C | 1.059 (0.998–1.119) | Yes |
| | B/Phuket | Batch A vs. batch B | 1.050 (1.010–1.090) | Yes |

*(Continued)*

**Table 5.** (Continued)

| Age group | Strain | Comparison | GMT post-vaccination ratio (95% CI) | Equivalence |
|---|---|---|---|---|
| | | Batch A vs. batch C | 1.049 (1.006–1.091) | Yes |
| | | Batch B vs. batch C | 1.024 (0.980–1.067) | Yes |

Age 9–12 years: *n* = 45 for each QIV batch A, B, and C group.

Age 13–17 years: *n* = 44 for QIV batch A group, *n* = 43 for QIV batch B group, and *n* = 45 for QIV batch C group.

Age 18–40 years: *n* = 45 for each QIV batch A, B, and C group.

the QIV and TIV groups, respectively. Mild myalgia was the most systemic adverse reaction, which occurred in 13.6% and 9.6% of the subjects in the QIV and TIV groups, respectively.

## Discussion

This study was conducted on subjects aged 9–40 years, while a previous study by Dhamayanti et al. was conducted on infants to children aged 8 years [17]. The HI data showed a strong serological response for each of the shared influenza strains in the QIV and TIV groups and the percentage of subjects in the QIV group achieving a serum HI titer ≥40. In this study, in the age group of 9–40 years, the QIV induced comparable immune responses to TIV for A strains and the B lineage common to both QIV and TIV. The protectivity/seroprotection rate of QIV was defined as the percentage of participants with an HI titer ≥40 for A/California/7/2009 (X-179A) (H1N1) pdm09, A/Hong Kong/4801/2014 (X-263) (H3N2), B/Texas/2/2013, and B/Phuket/3073/2013 of QIV (99.5%; 99.5%; 93.1%; 99.0%) and TIV (98.5%; 97.8%; 91.9%; 91.1%). The immunogenicity based on seroprotection rates of a candidate QIV was not significantly different from TIV for shared vaccine strains ($p > 0.01$) and was significantly different

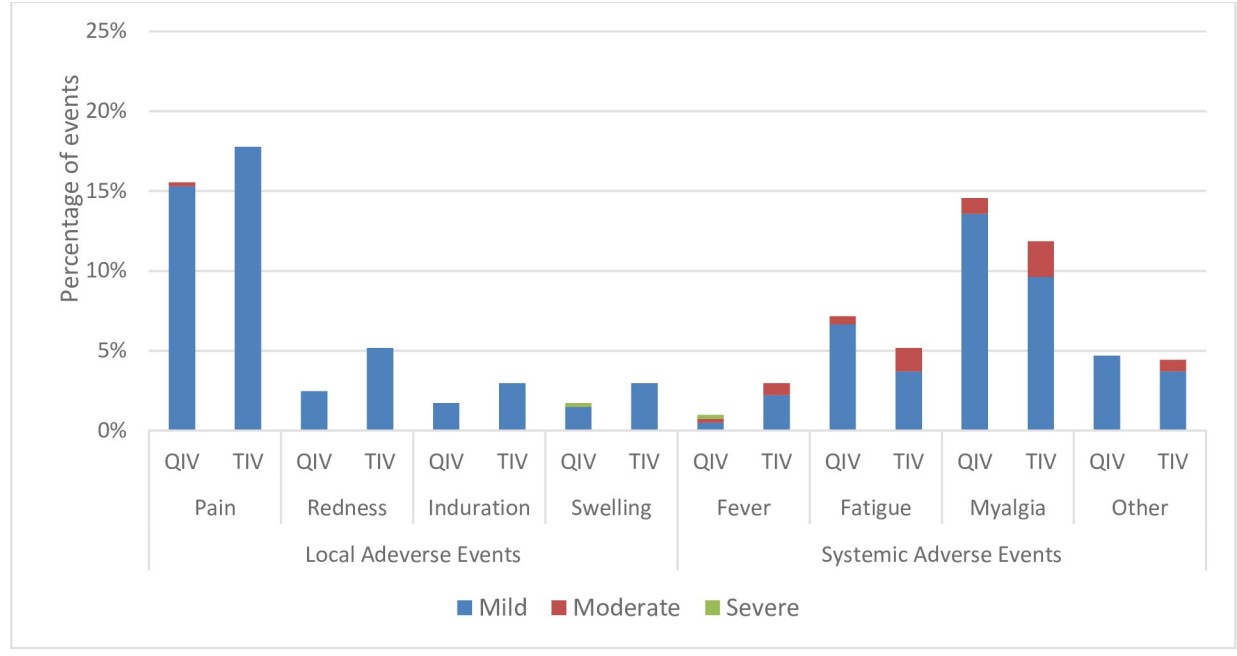

**Fig 3. Intensity of reported local and systemic adverse events (AEs).**

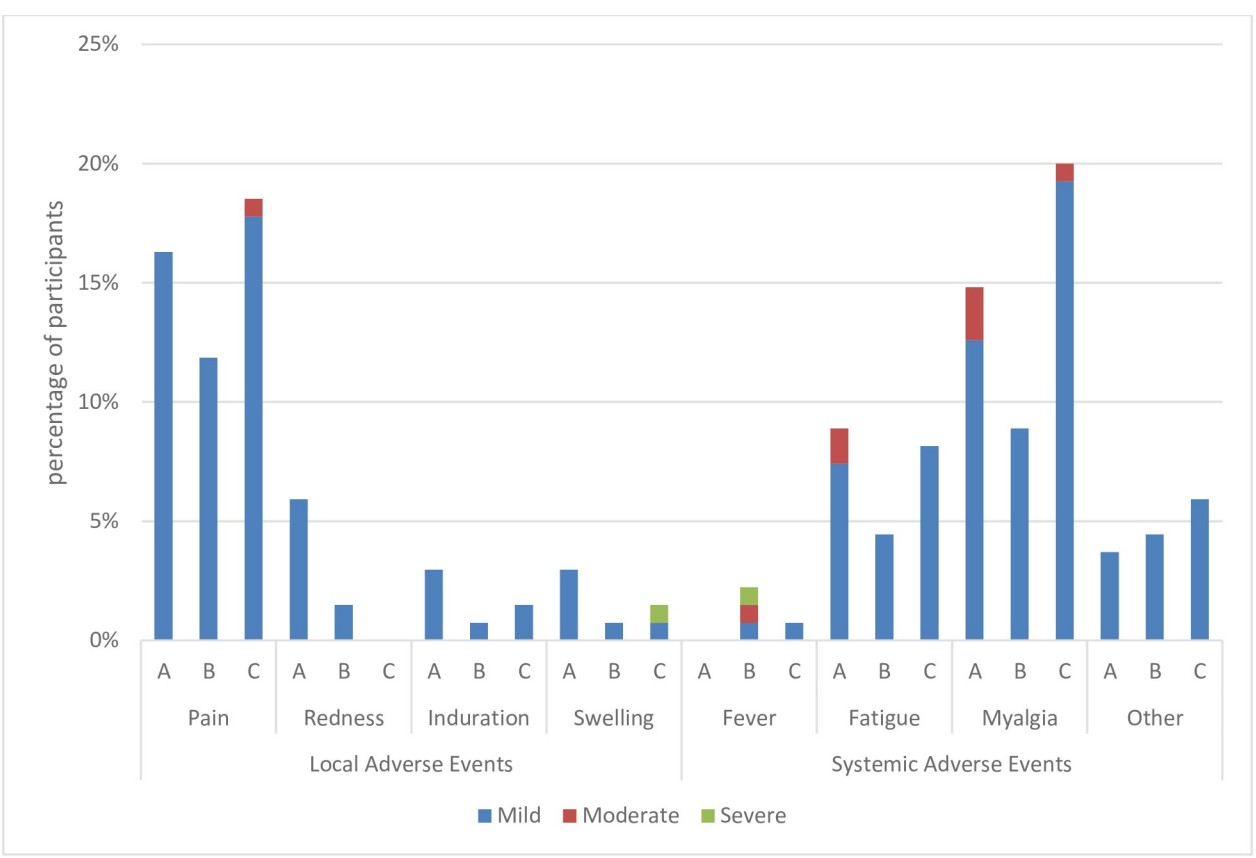

**Fig 4. Intensity of reported local and systemic AEs on each QIV batch.**

from TIV with respect to the added B strains (B/Phuket/3073/2013) ($p < 0.01$). The same result was observed for the GMT as well.

The inclusion of a fourth strain in QIV did not interfere with the EMA criteria for immune responses in adult vaccine recipients. In adults, post-vaccination seroprotection rates were ≥99%, seroconversion rates were >59%, and post-vaccination/pre-vaccination GMT ratios were ≥7.3 for all four vaccine strains. These data demonstrate that the presence of a second influenza B strain in QIV does not negatively affect the immune response to the other strains. Moreover, the immune responses to all strains contained in the two vaccines were robust, with the highest responses to the A/California/7/2009 (H1N1) strain.

The seroconversion rates regarding the percentage of subjects with increasing antibody titer ≥4 times and transition of seronegative to seropositive and of QIV were not significantly different from those of TIV. The superior QIV immunogenicity is expected to correspond with superior protection against influenza B relative to TIV in a season when there is lineage mismatch or cocirculation of two influenza B lineages. These findings are similar to those of the meta-analysis study by Moa in 2015, which confirmed that the inactivated QIV had similar efficacy against the three strains shared in common with the TIV (A/H1N1, A/H3N2, and the B lineage included in the TIV) [18]. The addition of a second influenza B strain in QIV might enhance the protection against influenza because it formulation adjustments could increase the effectiveness [19].

This study also showed that the immunogenicity of the three batches of QIV was equivalent for the four strains. Batch-to-batch equivalence of all three batches of QIV was demonstrated

for all four strains. The seroprotection rates of three batches for A/California/7/2009 (X-179A) (H1N1) pdm09, A/Hong Kong/4801/2014 (X-263) (H3N2), B/Texas/2/2013, and B/Phuket/3073/2013 were (99.3; 100; 99.3), (99.3; 99.2; 100), (88.8; 95.5; 94.8), and (98.5; 100; 98.5), respectively. The increase in the overall post-vaccination GMTs for each pair of batches for each strain was not different.

During this study, no serious AEs related to the vaccine were observed. This study found comparable reactogenicity and safety profiles between the QIV candidate and the TIV in both adult and adolescent groups. Both vaccines were well-tolerated by both age groups. Local and systemic reactogenicity profiles were also similar between the vaccine groups. Most reactions were mild or moderate in severity and lasted for 1–3 days. Most solicited injection-site and systemic reactions with either vaccine were mild to moderate and resolved within a few days.

Injection-site pain was the most frequently reported solicited local event, and fatigue and myalgia were the commonly reported solicited systemic events among the studies. A meta-analysis study by Moa in 2015 showed the same result, injection-pain was more common in QIV compared to TIV [18].

The QIV and TIV groups showed similar rates of systemic AEs. In both vaccine groups, mild myalgia was the most frequent systemic AE. These findings are similar to those reported by Wang et al. [20]. The incidence of fever was similar with both vaccines, which is consistent with the results of a previous study [21].

Frequencies of unsolicited AEs in the 28 days following vaccination with QIV were similar between the adult and adolescent groups. These data are consistent with the results of a meta-analysis of five randomized clinical trials, demonstrating no significant difference between QIV and TIV in terms of the frequency of aggregated local and systemic AEs within 7 days after vaccination [18].

QIV and TIV have similar reactogenicity and AE profiles, with no apparent adverse effects on the tolerability of the higher antigen content in QIV (60 μg HA for 4 strains compared with 22.5 μg for 3 strains in the TIV). Furthermore, the safety profiles of the two vaccine groups were comparable. The results of this study demonstrated that the additional B strain in QIV did not compromise safety compared with TIV. These findings are similar to those of the meta-analysis, which were no statistically significant differences between the adverse events in QIV and TIV groups [18].

In this study, immunogenicity was only assessed 28 days after the last vaccination. Hence, we do not know the duration of antibody responses to the 4 strains. Further study is needed to evaluate the antibody persistence of the QIV.

## Conclusion

In conclusion, this study demonstrated that QIV can be reproducibly manufactured to yield a well-tolerated, safe, and immunogenic vaccine in people aged 9–40 years and that it met all EMA immunogenicity criteria in adults. In adults, inactivated QIV induced comparable immune responses to TIV for A strains and the B lineage common to both QIV and TIV.

These data support the use of Bio Farma QIV for seasonal vaccination in children and adult subjects, which may enhance the protection against influenza and decrease the burden associated with influenza complications. The immunogenicity of the three batches of QIV was equivalent for all four strains.

## Supporting information

**S1 Checklist. CONSORT 2010 checklist of information to include when reporting a randomised trial*.**
(DOC)

**S1 File.**
(PDF)

## Acknowledgments

We would like to thank Dr. Iskandar, the Director of Bio Farma, for supporting this study. Thanks to Dr. Novilia S. Bachtiar, dr., M.Kes who has passed away for her contribution during various stages of the paper preparation. We would also like to thank the participants, Rita Verita Sri Hasniarty, the Head of Bandung District Health Office, Nitta Kurniati, the Head of Garuda Primary Health Center, and her staff, Siti Nurhasijatiningsih, the Head of Ibrahim Adjie Primary Health Center, and her staff, and Sylvie Virgianti, the Head of Puter Primary Health Center, and her staff for their work in this study. We also thank Mr. Hadyana Sukandar for his statistical work in this study and acknowledge the Indonesian National AEFI Committee as the auditor of serious AEs in this study. We would also like to express our appreciation to the staff at the Clinical Research Unit of Growth Development-Social Pediatric Division for the invaluable administrative assistance.

## Author Contributions

**Conceptualization:** Eddy Fadlyana, Meita Dhamayanti, Rodman Tarigan, Rini Mulia Sari, Kusnandi Rusmil, Cissy B. Kartasasmita.

**Data curation:** Eddy Fadlyana, Meita Dhamayanti, Rodman Tarigan, Rini Mulia Sari, Kusnandi Rusmil.

**Formal analysis:** Eddy Fadlyana, Meita Dhamayanti, Rodman Tarigan, Kusnandi Rusmil.

**Funding acquisition:** Eddy Fadlyana, Rini Mulia Sari, Kusnandi Rusmil.

**Investigation:** Eddy Fadlyana, Meita Dhamayanti, Rodman Tarigan, Kusnandi Rusmil.

**Methodology:** Eddy Fadlyana, Meita Dhamayanti, Rini Mulia Sari, Kusnandi Rusmil, Cissy B. Kartasasmita.

**Project administration:** Eddy Fadlyana, Rini Mulia Sari, Kusnandi Rusmil.

**Resources:** Eddy Fadlyana, Meita Dhamayanti, Rodman Tarigan, Rini Mulia Sari, Kusnandi Rusmil.

**Supervision:** Eddy Fadlyana, Meita Dhamayanti, Rini Mulia Sari, Kusnandi Rusmil.

**Validation:** Rini Mulia Sari, Kusnandi Rusmil.

**Visualization:** Eddy Fadlyana.

**Writing – original draft:** Eddy Fadlyana.

**Writing – review & editing:** Eddy Fadlyana, Meita Dhamayanti, Rodman Tarigan, Susantina Prodjosoewojo, Andri Reza Rahmadi, Rini Mulia Sari, Kusnandi Rusmil, Cissy B. Kartasasmita.

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
