## [Decision Letter · Decision Letter 0]

18 Apr 2023

PONE-D-22-35728Immunogenicity and safety of Quadrivalent Influenza HA vaccine compared with Trivalent Influenza HA vaccine and evaluation of Quadrivalent Influenza HA vaccine batch-to-batch consistency in Indonesian children and adultsPLOS ONE

Dear Dr. Fadlyana,

Thank you for submitting your manuscript to PLOS ONE. After careful consideration, we feel that it has merit but does not fully meet PLOS ONE’s publication criteria as it currently stands. Therefore, we invite you to submit a revised version of the manuscript that addresses the points raised below during the review process.

We look forward to receiving your revised manuscript.

Kind regards,

Ray Borrow, Ph.D., FRCPath

Academic Editor

PLOS ONE

Journal Requirements:

I have read the journal's policy and the authors of this manuscript have the following competing interests: Novilia Sjafri Bachtiar and Rini Mulia Sari are employees of PT Bio Farma at the time of the conduct of this study and manuscript preparation.   

We note that you received funding from a commercial source: PT Bio Farma

6. We note that the original protocol that you have uploaded as a Supporting Information file contains an institutional logo. As this logo is likely copyrighted, we ask that you please remove it from this file and upload an updated version upon resubmission.

7. We note that the original protocol file you uploaded contains a confidentiality notice indicating that the protocol may not be shared publicly or be published. Please note, however, that the PLOS Editorial Policy requires that the original protocol be published alongside your manuscript in the event of acceptance. Please note that should your paper be accepted, all content including the protocol will be published under the Creative Commons Attribution (CC BY) 4.0 license, which means that it will be freely available online, and any third party is permitted to access, download, copy, distribute, and use these materials in any way, even commercially, with proper attribution.

Therefore, we ask that you please seek permission from the study sponsor or body imposing the restriction on sharing this document to publish this protocol under CC BY 4.0 if your work is accepted. We kindly ask that you upload a formal statement signed by an institutional representative clarifying whether you will be able to comply with this policy. Additionally, please upload a clean copy of the protocol with the confidentiality notice (and any copyrighted institutional logos or signatures) removed.

Reviewers' comments:

Reviewer's Responses to Questions

**Comments to the Author**

1. Is the manuscript technically sound, and do the data support the conclusions?

Reviewer #1: Partly

Reviewer #2: Yes

2. Has the statistical analysis been performed appropriately and rigorously? 

Reviewer #1: No

Reviewer #2: Yes

3. Have the authors made all data underlying the findings in their manuscript fully available?

Reviewer #1: Yes

Reviewer #2: Yes

4. Is the manuscript presented in an intelligible fashion and written in standard English?

Reviewer #1: Yes

Reviewer #2: Yes

5. Review Comments to the Author

Reviewer #1: This manuscript reports immunogenicity and safety of Quadrivalent Influenza HA vaccine compared with Trivalent Influenza HA vaccine and evaluation of Quadrivalent Influenza HA vaccine batch-to-batch consistency in Indonesian children and adults. I have below comments.

Page 5, line 91, “open-labeled” is not consistent with the description in trial protocol where “double-blind” was panned for 9-40 years of age.

Page 6 sample size consideration is not consistent with the description in trial protocol. Please correct and make the descriptions in details about how sample sizes were determined.

Page 7, line 129, (1/1-0.1)*115 cannot obtain 128.

Page 7, line 133, please make it clear which data analysis will use these mentioned statistical tests.

Page 9, Table 2, Mann-Whitney test is not suitable to test count or proportion data. Chi-square may be used.

Please tell what statistical test was used for Fig. 2.

Reviewer #2: This was an open-labeled, bridging clinical study to assess the immunogenicity and safety of QIV compared with TIV and to evaluate batch-to-batch consistency in three consecutive batches of QIV. As a solid, well-performed clinical trial, this manuscript will be an important addition to the literature, however, there are some points that the authors should address that can improve the manuscript.

Major comments

1. The title of this manuscript includes “evaluation of Quadrivalent Influenza HA vaccine batch-to-batch consistency in Indonesian children and adults”, however, the results of this part are not described in the conclusion of the abstract.(Page3, Line38-40)

2. “This was the first QIV study conducted by Bio Farma on subjects aged 9-40 years in Indonesia.”, the sentence here does not connect the context and the reference here is the first study on QIV, which is a supplement to QIA study. I think it should be described in detail in the text to increase the rigor of the article.(Page5, Line83-84)

3. The content about study design is too concise to figure out, this part should be expanded.(Page5, Line89)

4. Please also fully list inclusion/exclusion criteria in the main text, and confirm these have been unchanged from the protocol.(Page5, Line101-106)

5. Please state clearly, in the methods what were the primary and secondary endpoints of the study.(Page6, Line125)

6. Table 4 as the main table should be described in the text.(Page11, Line183)

7. If the p value is <0.0001, please indicate it in the figure.(Figure 2)

Minor comments

1. Materials and methods: “Sample size and study analysis” is best described in two parts.(Page6, Line125)

2. Line 101-106, please carefully check the inclusion and exclusion criteria are consistent with the protocol.

3. Line 146, the results of this manuscript are poorly described, and the author is advised to restate it. In particular, important results should be detailed.

4. In figure 1, QIV Batch A, “exluded from analysis (Protocol noncompliant, n=134)” should be “excluded” and “n=1”.

6. PLOS authors have the option to publish the peer review history of their article (what does this mean?). If published, this will include your full peer review and any attached files.

Reviewer #1: No

Reviewer #2: No

---

## [Author Response · Author response to Decision Letter 0]

18 May 2023

We appreciate the editorial staff and the reviewers for taking time to carefully review the manuscript and give detailed and constructive comments, which has greatly helped to improve this paper. Below is our point-by-point response to each comment.

Journal Requirements:

Response : Thank you reminding us. We have made sure that the manuscript meets the PLOS ONE’s style requirements. 

I have read the journal's policy and the authors of this manuscript have the following competing interests: Novilia Sjafri Bachtiar and Rini Mulia Sari are employees of PT Bio Farma at the time of the conduct of this study and manuscript preparation. 

We note that you received funding from a commercial source: PT Bio Farma

Response : Thanks for the suggestions. We have added the following statement and revised the competin interests section. (Page 16, Line 338-342)

3. PLOS requires an ORCID ID for the corresponding author in Editorial Manager on papers submitted after December 6th, 2016. Please ensure that you have an ORCID iD and that it is validated in Editorial Manager. To do this, go to ‘Update my Information’ (in the upper left-hand corner of the main menu), and click on the Fetch/Validate link next to the ORCID field. This will take you to the ORCID site and allow you to create a new iD or authenticate a pre-existing iD in Editorial Manager. Please see the following video for instructions on linking an ORCID iD to your Editorial Manager account: https://www.youtube.com/watch?v=_xcclfuvtxQ

Response : Thank you for the reminding us. We have updated an ORCID iD to “Update my Information” 

Response : Thank you for pointing this out. We have made sure the ethics statement only appear in the Methods section.

Response : Thank you for pointing this out. We have added the caption of our Supporting Information files at the end of the manuscript. (Page 20, Line 418-423)

6. We note that the original protocol that you have uploaded as a Supporting Information file contains an institutional logo. As this logo is likely copyrighted, we ask that you please remove it from this file and upload an updated version upon resubmission.

Response : Thank you for reminding us. We have removed the institutional logo from the original protocol.

7. We note that the original protocol file you uploaded contains a confidentiality notice indicating that the protocol may not be shared publicly or be published. Please note, however, that the PLOS Editorial Policy requires that the original protocol be published alongside your manuscript in the event of acceptance. Please note that should your paper be accepted, all content including the protocol will be published under the Creative Commons Attribution (CC BY) 4.0 license, which means that it will be freely available online, and any third party is permitted to access, download, copy, distribute, and use these materials in any way, even commercially, with proper attribution.

Therefore, we ask that you please seek permission from the study sponsor or body imposing the restriction on sharing this document to publish this protocol under CC BY 4.0 if your work is accepted. We kindly ask that you upload a formal statement signed by an institutional representative clarifying whether you will be able to comply with this policy. Additionally, please upload a clean copy of the protocol with the confidentiality notice (and any copyrighted institutional logos or signatures) removed.

Response : We have attached the the study sponsor’s permission along the other supporting file.

Review Comments to the Author

Reviewer #1: This manuscript reports immunogenicity and safety of Quadrivalent Influenza HA vaccine compared with Trivalent Influenza HA vaccine and evaluation of Quadrivalent Influenza HA vaccine batch-to-batch consistency in Indonesian children and adults. I have below comments.

Page 5, line 91, “open-labeled” is not consistent with the description in trial protocol where “double-blind” was panned for 9-40 years of age.

Response : Thank you for reminding us. We have revised this part based on the protocol.

Revised manuscript (Page 2, Line 24)

This was an experimental, randomized, double blind, four arm parallel group bridging study involving unprimed healthy children and adults aged 9–40 years. A total of 540 subjects were enrolled in this study and randomized into four arm groups.

Revised manuscript (Page 5, Line 96-98)

This was an experimental, randomized, double blind, four arm parallel group bridging study to assess immune response and safety after one dose of quadrivalent and trivalent influenza HA vaccine in subjects 9-40 years old in clinically health children and adults aged 9–40 years.

Page 6 sample size consideration is not consistent with the description in trial protocol. Please correct and make the descriptions in details about how sample sizes were determined.

Response : Thank you for the suggestion. We have revised the descriptions about how sample sizes were determined clearly.

Revised manuscript (Page 7, Line 140-144)

The sample size was determined based on a 95% confidence interval (CI) and a test power of 80%. The required sample size was 112 in each batches, with the assumption that not all the subjects could complete the study, the total number of subjects was added at least 20% from the minimum requirement (N x 1.2) = 134. Approximately 135 subjects per group will be involved in this study. 

Page 7, line 129, (1/1-0.1)*115 cannot obtain 128.

Response : Thank you for pointing this out. We have revised this part.

Revised manuscript (Page 7, Line 141-144)

The required sample size was 112 in each batches, with the assumption that not all the subjects could complete the study, the total number of subjects was added at least 20% from the minimum requirement (N x 1.2) = 134. Approximately 135 subjects per group will be involved in this study.

Page 7, line 133, please make it clear which data analysis will use these mentioned statistical tests.

Response : We have revised this part to make it clear.

Revised manuscript (Page 7, Line 148-153)

Demographic data were expressed as mean, standard deviation, and range values. Analysis of Geometric Mean Titer (GMT), seroprotection, and seroconversion rates between the vaccine groups was performed using Kruskal-Wallis and Chi-square or Kolmogorov-Smirnov tests. The differences of antibody concentration for each antigen before and after primary series of Influenza vaccine was analysed using Wilcoxon test. Values of p < 0.05 indicated statistically significant differences between groups were assessed by Chi-square test. Immunogenicity analyses were performed on the per protocol population. Safety data were analysed in the intention to treat population.

Page 9, Table 2, Mann-Whitney test is not suitable to test count or proportion data. Chi-square may be used.

Response : We apologize for the mistake. For table 2, we used chi-square test for analyzed the data and we have revised the explanation. (Table 2; Page 9, Line 185-186)

Please tell what statistical test was used for Fig. 2.

Response : For figure 2, we used mann whitney test as the statistical test (Figure 2)

Reviewer #2: This was an open-labeled, bridging clinical study to assess the immunogenicity and safety of QIV compared with TIV and to evaluate batch-to-batch consistency in three consecutive batches of QIV. As a solid, well-performed clinical trial, this manuscript will be an important addition to the literature, however, there are some points that the authors should address that can improve the manuscript.

Major comments

1. The title of this manuscript includes “evaluation of Quadrivalent Influenza HA vaccine batch-to-batch consistency in Indonesian children and adults”, however, the results of this part are not described in the conclusion of the abstract.(Page3, Line38-40)

Response : Thank you for the comments. We have added the results of the “methods and findings” part in the abstract to describe the title. And in the “conclusion” of the abstract, there has been already a part which describes the title. 

Revised manuscript (Page 2, Line 38-39)

This was an open-labeled, bridging clinical study involving unprimed healthy children and adults aged 9–40 years. A total of 540 subjects were enrolled in this study and randomized into four arm groups. Each subject received one dose of TIV or QIV with three different batch codes. Serology tests were performed at baseline and 28 days after vaccination. Hemagglutination inhibition (HI) antibody titers were analyzed for Geometric Mean Titer (GMT), seroprotection, and seroconversion rates. Solicited, unsolicited, and serious adverse events were observed up to 28 days after vaccination. A total of 537 subjects completed the study per protocol and were analyzed for immunogenicity criteria. All randomized subjects were analyzed for safety criteria. The percentage of the subjects with anti-HI titer ≥1:40 28 days after QIV vaccination was 99.5% for A/H1N1; 99.5% for A/H3N2; 93.1% for B/Texas, and 99.0% for B/Phuket. The seroprotection, GMT, and seroconversion rates of QIV were not significantly different from those of TIV for the common vaccine strains (p > 0.01) and were significantly different from those of TIV for the added B/Phuket strains (p < 0.01). Most solicited injection-site and systemic reactions with either vaccine were mild to moderate and resolved within a few days. Antibody response to QIV were equivalence among vaccine batches and comparable between age groups for each of the 4 strains.

2. “This was the first QIV study conducted by Bio Farma on subjects aged 9-40 years in Indonesia.”, the sentence here does not connect the context and the reference here is the first study on QIV, which is a supplement to QIA study. I think it should be described in detail in the text to increase the rigor of the article.(Page5, Line83-84)

Response : Thank you for pointin out this term. We have revised the sentences in this paraghraph. 

Revised manuscript (Page 5, Line 84-90)

At first, Bio Farma formulated TIVs and have conducted several studies on seasonal TIVs between 2008 and 2014. The results of these studies were consistent. TIVs were well-tolerated and induced high antibody titer against influenza antigens and no serious adverse events (AEs) during the study [9–13]. In 2016, Bio Farma starts to formulate the quadrivalent inactivated influenza vaccine that could potentially provide wider protection against influenza B viruses. Several countries has been conducted the studies using Influenza HA Vaccine (quadrivalent vaccine), but none from Indonesia [8,14,15]. This was the first QIV study conducted by Bio Farma on subjects aged 9–40 years in Indonesia .

3. The content about study design is too concise to figure out, this part should be expanded.(Page5, Line89)

Response : We have expanded the explanation of study design part. (Page 5, Line 95-104)

4. Please also fully list inclusion/exclusion criteria in the main text, and confirm these have been unchanged from the protocol.(Page5, Line101-106)

Response : Thank you for the suggestions. We have added the inclusion and exclusion criteria in main text based on the protocol 

Revised manuscript (Page 6, Line 108-120)

The primary inclusion criteria included healthy children and adults aged 9–40 years who have signed the informed consent form and committed to complying with study instructions and trial schedules. Subjects were not eligible if the subject have enrolled or scheduled to be enrolled in another trial, presented with mild, moderate, or severe illness with a fever (axillary temperature ≥37.5⁰C), history of allergy to egg, chicken protein, or other vaccine, uncontrolled coagulopathy or blood disorders contraindicating intramuscular injection, received in the previous 4 weeks a treatment likely to alter the immune response [intravenous immunoglobulins, blood-derived products, or long-term corticosteroid therapy (>2 weeks)], pregnancy and lactation (adult), have any abnormality or chronic disease which according to the investigator might interfere with the assessment of the trial objectives, have already immunized with influenza vaccine within 1 year or any vaccination within 1 month before and after immunization of Quadrivalent Influenza Vaccine.

5. Please state clearly, in the methods what were the primary and secondary endpoints of the study.(Page6, Line125)

Response : Thank you for the suggestions. We have added the primary and secondary endpoints in “Outcomes” part in Methods section. 

Revised manuscript (Page 8, Line 165-171)

The primary outcome was to evaluate the percentage of subjects with anti-HI titer � 1:40, 28 days after Influeza HA vaccination. The secondary outcome was to evaluate geometric mean titres, percentage of subjects with increasing antibody titer � 4 times and/or percentage of subjects with transition of seronegative to seropositive between one dose of quadrivalent and trivalent influenza HA vaccine in subjects 9-40 years old and between each batch number of Quadrivalent Influenza HA vaccine and the incidence rate and intensity of adverse events or any serious adverse events within 30 minutes, 72 hour and 28 days after immunization.

6. Table 4 as the main table should be described in the text.(Page11, Line183)

Response : Thank you for reminding us. We have added the explanation of table 4 in the text.

Revised manuscript (Page 10, Line 213-216)

Batch-to-batch equivalence for each strain was concluded if the two-sided 95% CI of each strain GMT ratio of the compared batches was between 0.67 and 1.5 (Table 4) [14]. The GMT ratios of the overall post-vaccination GMTs for each pair of lots for each strain were all between 0.67 and 1.5.

7. If the p value is <0.0001, please indicate it in the figure.(Figure 2)

Response : Thank you for the suggestion. We have added the p value (<0.001) in figure 2 (day 28, B/Phuket)

Minor comments

1. Materials and methods: “Sample size and study analysis” is best described in two parts.(Page6, Line125)

Response : Thank you for the suggestions. We have described those parts separately. (Sample Size; Page 7, Line 139-144 and Study analysis; Page 7-8, Line 145-171)

2. Line 101-106, please carefully check the inclusion and exclusion criteria are consistent with the protocol.

Response : Thank you for pointing this out. We have added the inclusion and exclusion criteria based on the protocol.

Revised manuscript (Page 6, Line 108-120)

The primary inclusion criteria included healthy children and adults aged 9–40 years who have signed the informed consent form and committed to complying with study instructions and trial schedules. Subjects were not eligible if the subject have enrolled or scheduled to be enrolled in another trial, presented with mild, moderate, or severe illness with a fever (axillary temperature ≥37.5⁰C), history of allergy to egg, chicken protein, or other vaccine, uncontrolled coagulopathy or blood disorders contraindicating intramuscular injection, received in the previous 4 weeks a treatment likely to alter the immune response [intravenous immunoglobulins, blood-derived products, or long-term corticosteroid therapy (>2 weeks)], pregnancy and lactation (adult), have any abnormality or chronic disease which according to the investigator might interfere with the assessment of the trial objectives, have already immunized with influenza vaccine within 1 year or any vaccination within 1 month before and after immunization of Quadrivalent Influenza Vaccine.

3. Line 146, the results of this manuscript are poorly described, and the author is advised to restate it. In particular, important results should be detailed.

Response : Thanks to the reviewer for pointing out this part. We have re-described the results section and have pointed out the important results. (Line 173)

4. In figure 1, QIV Batch A, “exluded from analysis (Protocol noncompliant, n=134)” should be “excluded” and “n=1”.

Response : Thanks for the comment. We have revised the description in figure 1. (Figure 1)

6. PLOS authors have the option to publish the peer review history of their article (what does this mean?). If published, this will include your full peer review and any attached files.

Response : Thanks for the chance. We agree to publish the peer review history for our manuscript.

Response : Thanks to the reviewer for pointing out this part. We already converted Figure Files to PACE.

---

## [Decision Letter · Decision Letter 1]

31 May 2023

PONE-D-22-35728R1Immunogenicity and safety of Quadrivalent Influenza HA vaccine compared with Trivalent Influenza HA vaccine and evaluation of Quadrivalent Influenza HA vaccine batch-to-batch consistency in Indonesian children and adultsPLOS ONE

Dear Dr. Fadlyana,

Thank you for submitting your manuscript to PLOS ONE. After careful consideration, we feel that it has merit but does not fully meet PLOS ONE’s publication criteria as it currently stands. Therefore, we invite you to submit a revised version of the manuscript that addresses the last remaining points below raised during the review process.

We look forward to receiving your revised manuscript.

Kind regards,

Ray Borrow, Ph.D., FRCPath

Academic Editor

PLOS ONE

Journal Requirements:

Reviewers' comments:

Reviewer's Responses to Questions

**Comments to the Author**

1. If the authors have adequately addressed your comments raised in a previous round of review and you feel that this manuscript is now acceptable for publication, you may indicate that here to bypass the “Comments to the Author” section, enter your conflict of interest statement in the “Confidential to Editor” section, and submit your "Accept" recommendation.

Reviewer #1: (No Response)

2. Is the manuscript technically sound, and do the data support the conclusions?

Reviewer #1: (No Response)

3. Has the statistical analysis been performed appropriately and rigorously? 

Reviewer #1: (No Response)

4. Have the authors made all data underlying the findings in their manuscript fully available?

Reviewer #1: (No Response)

5. Is the manuscript presented in an intelligible fashion and written in standard English?

Reviewer #1: (No Response)

6. Review Comments to the Author

Reviewer #1: Line 149-150, “The 149 differences of antibody concentration for each antigen before and after primary series of Influenza vaccine was analysed using Wilcoxon test”. Please indicate clearly which Wilcoxon test was used.

Line 150-151, “Values of p < 0.05 indicated statistically significant differences between groups were assessed by Chi-square test”. This sentence is not clear. Why was this cut-off p-value only used for analysis assessed by Chi-square test? You may want to say “Values of p < 0.05 indicated statistically significant differences between groups.”

7. PLOS authors have the option to publish the peer review history of their article (what does this mean?). If published, this will include your full peer review and any attached files.

Reviewer #1: No

---

## [Author Response · Author response to Decision Letter 1]

15 Jul 2023

PONE-D-22-35728R1

Immunogenicity and safety of Quadrivalent Influenza HA vaccine compared with Trivalent Influenza HA vaccine and evaluation of Quadrivalent Influenza HA vaccine batch-to-batch consistency in Indonesian children and adults

PLOS ONE

Dear editorial staff and the reviewers,

We appreciate the editorial staff and the reviewers for taking time to carefully review the manuscript and give detailed and constructive comments, which has greatly helped to improve this paper. Below is our point-by-point response to each comment.

Journal Requirements:

Response : Thank you for reminding us. We have reviewed our reference list, and we have changed some cite and references which couldn’t be accessed anymore to the new one.

Reference no. 7

World Health Ogranization. Recommended composition of influenza virus vaccines for use in the 2013-2014 northern hemisphere influenza season. February 2013. Available from: https://www.who.int/publications/m/item/recommended-composition-of-influenza-virus-vaccines-for-use-in-the-2013-2014-northern-hemisphere-influenza-season (accessed June 12, 2023).

Reference no. 13

Luo FJ, Yang LQ, Ai X, Bai YH, Wu J, Li SM, et al. Immunogenicity and safety of three 2010-2011 seasonal trivalent influenza vaccines in Chinese toddlers, children and older adults. Hum Vaccin Immunother. 2013 Aug 1; 9(8): 1725–1734. doi: 10.4161/hv.24832.

Reference no. 16

Food and Drug Administration Center. Clinical Data Needed to Support the Licensure of Seasonal Inactivated Influenza Vaccines. Guidance for Industry. 2007: 1-16. Available from: https://www.fda.gov/regulatory-information/search-fda-guidance-documents/clinical-data-needed-support-licensure-seasonal-inactivated-influenza-vaccines. (accessed June 12, 2023)

Comments to the Author

1. If the authors have adequately addressed your comments raised in a previous round of review and you feel that this manuscript is now acceptable for publication, you may indicate that here to bypass the “Comments to the Author” section, enter your conflict of interest statement in the “Confidential to Editor” section, and submit your "Accept" recommendation.

Response : Thank you for the comment. We have adequately addressed our comments raised in the previous round of review. 

2. Is the manuscript technically sound, and do the data support the conclusions?

Response : Yes, our manuscript has already described a technically sound and the data has already supported the conclusions

3. Has the statistical analysis been performed appropriately and rigorously? 

Response : Yes, the statistical analysis has been already performed appropriately and rigorously. 

4. Have the authors made all data underlying the findings in their manuscript fully available?

Response : Thank you for reminding us. We have added one table (Table 3. Geometric Mean Titers (GMT) to influenza A (H1N1), A (H3N2), and B virus strains Pre- and 28 days post-immunization) to explain the result in page 9, line 187-192.

Revised manuscript (Page 9, line 185-190)

Comparison of GMT between subjects who received QIV and TIV was shown on table 3 and figure 2. No significant differences in GMT 28 days after immunization between one dose of QIV and TIV in the 9–40-year-old subjects for A/California/7/2009 (X-179A) (H1N1) pdm09, A/Hong Kong/4801/2014 (X-263) (H3N2), and B/Texas/2/2013 (p = 0.322, p = 0.536, and p = 0.378, respectively). However, in strain B/Phuket/3073/2013, GMT 28 days after immunization was significantly higher in QIV group than in TIV group (p < 0.001).

Table 3. Geometric Mean Titers (GMT) to influenza A (H1N1), A (H3N2), and B virus strains Pre- and 28 days post-immunization

Fig 2. Hemagglutination inhibition (HI) Geometric Mean Titers (GMT) before and 28 days after QIV and TIV immunization.

5. Is the manuscript presented in an intelligible fashion and written in standard English?

Response : We have already made sure that language we used in this manuscript have been clear, correct and unambiguous.

6. Review Comments to the Author

Reviewer #1: Line 149-150, “The 149 differences of antibody concentration for each antigen before and after primary series of Influenza vaccine was analysed using Wilcoxon test”. Please indicate clearly which Wilcoxon test was used.

Response : In this manuscript, there is no data which used Wilcoxon test to analyse the data. We have deleted this part. (Page 7, line 146-151)

Line 150-151, “Values of p < 0.05 indicated statistically significant differences between groups were assessed by Chi-square test”. This sentence is not clear. Why was this cut-off p-value only used for analysis assessed by Chi-square test? You may want to say “Values of p < 0.05 indicated statistically significant differences between groups.”

Response : Thank you for the suggestion. Yes, we have revised this line as suggested. (Page 7, line 149)

7. PLOS authors have the option to publish the peer review history of their article (what does this mean?). If published, this will include your full peer review and any attached files.

Do you want your identity to be public for this peer review? For information about this choice, including consent withdrawal, please see our Privacy Policy.

Reviewer #1: No

---

## [Editor Report · Decision Letter 2]

2 Aug 2023

Immunogenicity and safety of Quadrivalent Influenza HA vaccine compared with Trivalent Influenza HA vaccine and evaluation of Quadrivalent Influenza HA vaccine batch-to-batch consistency in Indonesian children and adults

PONE-D-22-35728R2

Dear Dr. Fadlyana,

We’re pleased to inform you that your manuscript has been judged scientifically suitable for publication and will be formally accepted for publication once it meets all outstanding technical requirements.

Kind regards,

Ray Borrow, Ph.D., FRCPath

Academic Editor

PLOS ONE
---

## [Editor Report · Acceptance letter]

15 Aug 2023

PONE-D-22-35728R2 

Immunogenicity and safety of Quadrivalent Influenza HA vaccine compared with Trivalent Influenza HA vaccine and evaluation of Quadrivalent Influenza HA vaccine batch-to-batch consistency in Indonesian children and adults 

Dear Dr. Fadlyana:

I'm pleased to inform you that your manuscript has been deemed suitable for publication in PLOS ONE. Congratulations! Your manuscript is now with our production department. 

Kind regards, 

on behalf of

Prof. Ray Borrow 

Academic Editor

PLOS ONE